# A Framework of Generating Land Surface Reflectance of China Early Landsat MSS Images by Visibility Data and Its Evaluation

Cong Zhao [1], Zihua Wu [1], Qiming Qin [1,2,*] and Xin Ye [1,2]

1    Institute of Remote Sensing and Geographical Information System, School of Earth and Space Sciences, Peking University, Beijing 100871, China; zhaocong2013@pku.edu.cn (C.Z.); wuzihua@pku.edu.cn (Z.W.); xinye@pku.edu.cn (X.Y.)

2    Beijing Key Laboratory of Spatial Information Integration and 3S Application, Peking University, Beijing 100871, China

\*    Correspondence: qmqin@pku.edu.cn

**Abstract:** The Landsat time-series dataset is one of the most widely used datasets for land surface research due to its long time-series and Land Surface Reflectance (LSR) product. Though the United States Geological Survey (USGS) provides Landsat LSR products for later Landsat 4–5 Thematic Mapper (TM), Landsat 7 Enhanced Thematic Mapper Plus (ETM+), and Landsat 8 Operational Land Imager (OLI), no early Landsat 1–5 Multispectral Scanner System (MSS) LSR product is generated currently, limiting the research traced back to the 1970s. Atmospheric correction is one of the necessary preprocesses for generating LSR products. However, it is challenging for MSS images, not only because the image quality is lower and bands are different compared with the current sensors, but also because of the multiple effects of other preprocesses, such as radiometric calibration. Based on the Second Simulation of a Satellite Signal in the Solar Spectrum Vector (6SV) model, we propose a novel framework for generating Landsat 1–5 MSS LSR data of China. Ground-based visibility records are introduced to replace the images-based aerosol optical depth (AOD) to effectively generate MSS LSR data of the 1970s. We evaluate the generated MSS LSR data by the cross-validation of the simultaneous observation of MSS and TM sensors in Landsat 4 and Landsat 5 using Landsat Ecosystem Disturbance Adaptive Processing System (LEDAPS) surface reflectance product as the truth value. The evaluation result shows that the generated MSS LSR data is comparable with the later Landsat TM LSR product, with slightly larger uncertainties. In addition, it shows that the non-atmospheric factors (e.g., the difference of relative spectral responses of TM and MSS, the georegistration errors, the radiometric calibration uncertainty, and image noises) bring larger uncertainties than the atmospheric factors (e.g., the AOD retrieval method by visibility) to the cross-validation results. We apply the MSS LSR data generated by the proposed framework on time series analysis in the regions of interest (ROIs) of the spectral-stable land cover in China for all the MSS sensors. The application demonstrates the potential and promise of the MSS LSR data generated by the proposed framework.

**Keywords:** early Landsat MSS; atmospheric correction; uncertainties analysis; time series

## 1. Introduction

    As one of the most successful remote sensing datasets, the Landsat time-series dataset is widely used in land surface research, thanks to its long time-series and products [1]. Using Landsat Ecosystem Disturbance Adaptive Processing System (LEDAPS) for TM/ETM+ of Landsat 4, 5, and 7 and Land Surface Reflectance Code (LaSRC) for OLI of Landsat 8, a global on-demand atmospherically corrected surface reflectance product capability is reached currently [2–4]. However, the Landsat 1–5 Multispectral Scanner System (MSS) images acquired from July 1972 are not as easy to use as the later Landsat data due to the challenges of radiometric calibration [5,6], georegistration [7,8], and atmospheric correction [9,10]. Since surface reflectance is always considered a necessary and minimum





standard for analysis-ready data among most research now [1], the lack of MSS land surface reflectance (LSR) data, especially the lack of atmospheric correction of MSS, limits the application of remote sensing extending to the 1970s, as the Landsat MSS dataset is the only multi-spectral global observation record using remote sensing in the 1970s.

In the early years, statistical methods for atmospheric correction were proposed for specific remote sensing images, such as invariant-object methods [11], histogram matching methods [12,13], and dark object subtraction methods [14,15]. These methods are commonly based on solid assumptions, sometimes considered in doubt [16,17]. Afterward, atmospheric correction methods were significantly improved by incorporating a physically-based procedure [18]. Atmospheric correction using a physically-based procedure can be divided into two problems, building radiative transfer (RT) models and estimating input parameters of RT models.

With the development of MODTRAN [19] and the 6S model [20], the difficulty of atmospheric correction lay in how to accurately measure or estimate aerosol optical depth (AOD), the most important factor to an optical remote sensing image. The most influential ground-based AOD monitor network is AERONET [21]. The observation records of AERONET extended back to the early 1990s, but it had been still two decades since Landsat MSS had begun Earth observation. On the other hand, satellite-based methods estimating AOD were proposed, known as the dense dark vegetation (DDV) method for MODIS and Landsat TM images [22–25]. The DDV method is based on the known dense vegetation properties, that the blue and red band reflectance of dense dark vegetation is assumed as $0.02 \pm 0.01$ and the green band reflectance is assumed as $0.03 \pm 0.01$ [22], or that the empirical relationships among reflectance of blue, green, and shortwave infrared (SWIR) bands are known [16,24,25]. Additionally, there must be enough dense pixels and corresponding bands in the remote sensing image. Limited to the different bands and lower quality of MSS sensors, no stable and widely-applied method for AOD retrieval based on MSS images is proposed currently.

Because AOD and visibility respectively represent the vertical and horizontal attenuation of atmosphere, the relationship between AOD and visibility was found and developed [26–28]. Therefore, both methods retrieving AOD from visibility [29,30] and methods retrieving visibility from AOD [31,32] are proposed. As is advantageous in stable long time series and numerous observation stations, visibility data are used for temporal–spatial change research of AOD [30,33]. However, little research has studied atmospheric correction for remote sensing images using visibility data.

In this paper, we propose a novel framework to generate Landsat 1–5 MSS LSR data of China, using the ground-based visibility observation network records as the input of the 6S model. Considering the MSS images were acquired about 30–50 years ago with rare simultaneous in situ LSR observation records, we studied the uncertainty of the MSS land surface reflectance, using simultaneous TM LEDAPS product of Landsat 4–5 as the truth value. We also selected several ROIs of spectral-stable land cover in China and analyzed the time series reflectance of the ROIs to examine the stability of the visibility-based surface reflectance in an application of the data generated by our method.

The rest of the paper is organized as follows. Section 2 introduces the data used in the atmospheric correction of MSS images. Section 3 details the framework. Section 4 shows and analyzes the results and the sources of uncertainty quantitatively. Section 5 represents an application on time-series surface reflectance in five ROIs with spectral-stable land cover. Discussions and conclusions are presented, respectively, in Sections 6 and 7.

## 2. Data

### 2.1. Landsat Collection 1 Archive

Landsat satellites started in 1972, with Landsat 1–3 carrying the MSS sensor and Landsat 4–5 carrying both MSS and TM sensors. The MSS sensor presents four-band remote sensing images, covering visible and near-infrared (NIR) bands at $57 \times 79$ m resolution (resampled to 60 m in the Landsat Collection 1 Archive) and 6-bit quantization.

In contrast, the TM sensor onboard Landsat 4–5 presents seven bands at 30 m resolution and 8-bit quantization, with different bandpass positions.The main differences between MSS and TM images of Landsat Collection 1 Archive are shown in Table 1. The relative spectral responses (RSRs) of MSS and TM are downloaded from the USGS website (https://landsat.usgs.gov/spectral-characteristics-viewer, accessed on 3 April 2022). Figure 1 shows that there are noticeable differences between the relative spectral response of MSS and TM, but the MSS sensors of Landsat 1–5 are slightly varied.

**Table 1.** The difference between MSS and TM images of Landsat Collection 1 Archive.

|  | **MSS** | **TM** |
| --- | --- | --- |
| Bands | Green, red, NIR1, NIR2 | Blue, green, red, NIR, SWIR1, TIR, SWIR2 |
| Spatial resolution | 57 × 79 m (resampled to 60 m) | 30 m |
| Radiometric resolution | 6 bit (resampled to 8 bit) | 8 bit |

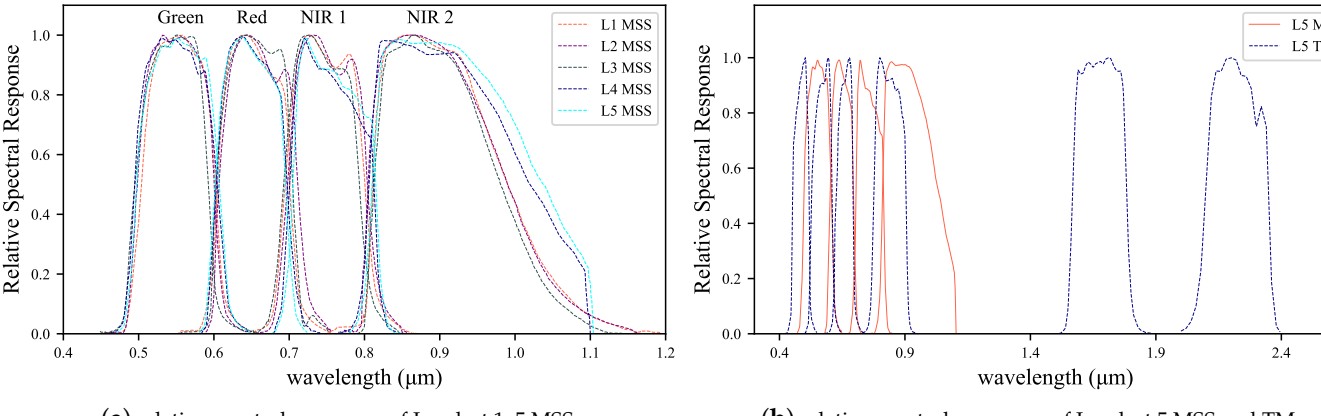

(**a**) relative spectral responses of Landsat 1–5 MSS   (**b**) relative spectral responses of Landsat 5 MSS and TM

**Figure 1.** Relative spectral responses of TM and MSS.

In 2016, the USGS reorganized the Landsat archive into a tiered collection management structure titled Landsat Collection 1 [1]. The Landsat 1–5 MSS data is resampled to 60 m resolution and 8-bit quantization and processed into Collection 1 through geometric registration [7] and radiometric calibration [6]. The geometric registration and radiometric calibration of MSS images are more difficult than later Landsat 4–5 TM images, Landsat 7 ETM+ images, and Landsat 8 OLI images. Only 0.61% of the MSS images can be processed to Tier 1, with tolerances of ≤12 m radial root mean square error (RMSE), while the percentages of TM, ETM+, and OLI are 59.63%, 70.04%, and 57.65% [34]. The absolute radiometric uncertainties of Landsat 1–5 MSS are 6–18%, while TM, ETM+, OLI are, respectively, 5–7%, 4%, and 3% [34].

The Landsat 4–5 satellites carried both MSS and TM sensors, which may simultaneously observe the Earth. The pair of MSS and TM images have the same observation time, target, and atmosphere condition, providing us cross-validation probability. In this research, we choose cloud-free images of Landsat 1–5 MSS Collection 1 in China during 1982–1995, with various land cover types, and we specially selected 479 pairs of MSS and TM images as cross-validation.

*2.2. Landsat 4–5 LEDAPS Surface Reflectance Product*

Landsat 4–5 LEDAPS Surface Reflectance Product is produced by the USGS on-demand processing system EROS Science Processing Architecture using LEDAPS. LEDAPS adopts additional water vapor, air pressure, and air temperature data from the National Centers for Environment Prediction (NCEP), ozone from the NASA Earth Probe Total

Ozone Mapping Spectrometer, and topography. LEDAPS uses the DDV method for AOD retrieval without auxiliary aerosols data [35].

The Landsat LEDAPS SR product is evaluated completely and thoroughly with good precision [2,3,36]. Using AERONET-AOD as true value, TM bands' average uncertainty (U) varies from 0.0041 (red band) to 0.0070 (SWIR-1 band) in reflectance, while TM bands' average uncertainty (U) varies from 0.009 (red band) to 0.017 (SWIR-1 band) in reflectance, using MODIS surface reflectance (after VJB Bidirectional Reflectance Distribution Function adjusted) as true value. As the MSS LSR data generated by our proposed framework are much more likely to be used with the LEDAPS product to form a long time series, we choose the LEDAPS SR product of Landsat TM as the truth value.

### 2.3. Integrated Surface Database

The Integrated Surface Database (ISD) consists of global hourly meteorological observations [37]. The earliest records of ISD can be traced back to the year 1901. Figure 2 shows the stations of ISD in 1973 in or near China.

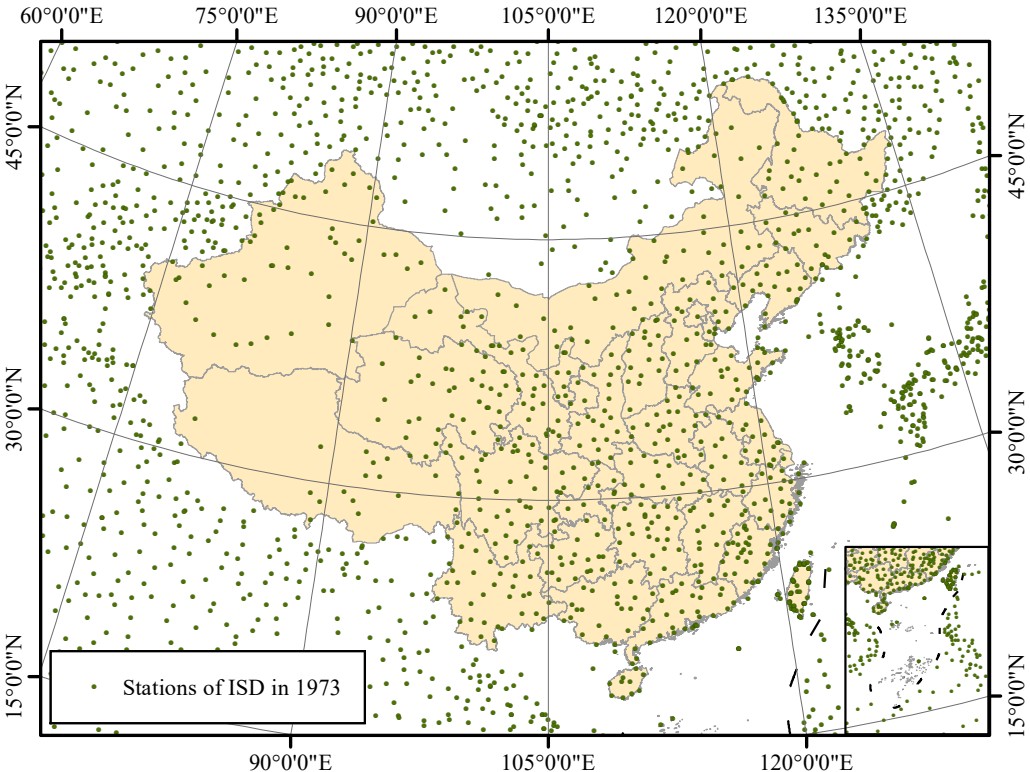

**Figure 2.** ISD stations in or near China in 1973.

ISD observes numerous phenomena, e.g., wind, temperature, pressure, visibility, and precipitation. The quality control algorithms integrated the data from over 100 original data sources and compiled them into ASCII format. In this research, we extract visibility records according to the image observation time and center coordinates by a spatial and temporal search radius.

### 2.4. NCEP Reanalysis

The NCEP/NCAR Reanalysis 1 is a widely-used dataset based on data assimilation [38]. The data are generated as 2.5 × 2.5 degrees global grids from 1948, covering air temperature, relative humidity, and specific humidity. This research uses daily specific humidity to calculate the column water vapor and surface air temperature and surface relative humidity to calculate surface water vapor pressure.

### 2.5. China Land Use/Cover Change (CNLUCC)

As the aerosol type is highly associated with land cover, we use China's Multi-Period Land Use Land Cover Remote Sensing Monitoring Dataset (CNLUCC) to determine the aerosol type of the MSS and TM images [39]. The CNLUCC dataset covers seven periods, including the late 1970s, late 1980s, mid-1990s, late 1990s, 2005, 2010, and 2015, and classifies the land cover into six classes and 25 subclasses by visual interpretation [39–41].

## 3. Methodology

### 3.1. Overview

The framework of generating the LSR of China's early MSS images by visibility data is shown in Figure 3. Similar to the later sensors, the framework includes the radiometric calibration to convert DN values to top-of-atmosphere (TOA) reflectance or radiance. We used the radiometric calibration coefficients of the metadata file (MTL.txt) provided by the Landsat Collection 1 Level 1 Archive, which was updated in 2019 [6]. Moreover, the observation date and time, geographic coordinates of the image center, and solar azimuth/elevation angle are all extracted from the metadata file.

The atmospheric RT model computed the transmission, intrinsic reflectance, and spherical albedo terms. Then, atmospheric correction coefficients are provided to transfer TOA radiance or reflectance to surface reflectance for every band. Since the visibility records are independent of the remote sensing observation, the quality assessment (QA) band is unnecessary, which helps extract the dense dark vegetation pixels for the DDV method.

The key of the framework is the module of retrieval and calculation of atmosphere parameters. The 6S model allows a direct visibility input, as an empirical model is built in the code. For contrast, we also applied a Qiu model [29], a physically-based model with an empirical correction function for AOD retrieval. The input parameters of empirical and physically-based models are different.

The visibility records are retrieved by search radiuses of 2 degrees spatially and 2 h temporally. All the visibility records observed two hours before and after the acquisition time within the 2 degrees area are retrieved. If the number of records retrieved is less than 4, the search radiuses increase with the maximum of 4 degrees spatially and 3 h temporally. The maximum visibility record of all the records is used, for the lower records are vulnerable to local environmental conditions. For those images with no visibility records retrieved, default visibility of 23 km is used. The content of WV is derived from NCEP/NCAR Reanalysis 1, and the ozone concentration is set as 0.345 atm-cm, as the Total Ozone Mapping Spectrometer (TOMS) data was not available during the 1970s. The aerosols model is precalculated by the land cover of China in 1980 and classified into four models, namely, continental, maritime, urban, and desert.

### 3.2. 6SV Atmospheric Correction Model

The 6SV radiative transfer model expresses the atmospheric correction equation in the Lambertian condition with no adjacency effects [4,20], as follows:

$$
\begin{aligned}
\rho_{TOA}&(\theta_s, \theta_v, P, Aer, U_{H_2O}, U_{O_3}) \\
&= Tg_{OG}(m, P)Tg_{O_3}(m, U_{O_3})[\rho_{atm}(\theta_s, \theta_v, \phi, P, Aer, U_{H_2 0}) \\
&+ Tr_{atm}(\theta_s, \theta_v, \phi, Aer)\frac{\rho_s}{1 - S_{atm}(P, Aer)\rho_s}Tg_{H_2 0}(m, U_{H_2O})]
\end{aligned}
\tag{1}
$$

where $\rho_{TOA}$ is the top-of-atmosphere (TOA) reflectance, $\rho_{atm}$ is the atmosphere intrinsic reflectance, $\rho_s$ is the land surface reflectance.

$Tr_{atm}$ is the total atmosphere transmission effected by AOD and water vapor (WV), $S_{atm}$ is the atmosphere spherical albedo, and $Tg$ is the gaseous transmission of absorbing gases in the atmosphere, including water vapor ($Tg_{wv}$), ozone ($Tg_{O_3}$), and other gases ($Tg_{og}$).

$\theta_v$, $\theta_s$, and $\phi$ are the view zenith angle, solar zenith angle, and relative azimuth angle, $m$ equals $1/cos(\theta_s) + 1/cos(\theta_v)$,

$P$ is the pressure that influences the number of molecules and the concentration, $U$ is the integrated content, $U_{H_2O}$ and $U_{O_3}$ for water vapor and ozone, respectively, and $Aer$ is the parameters of aerosol.

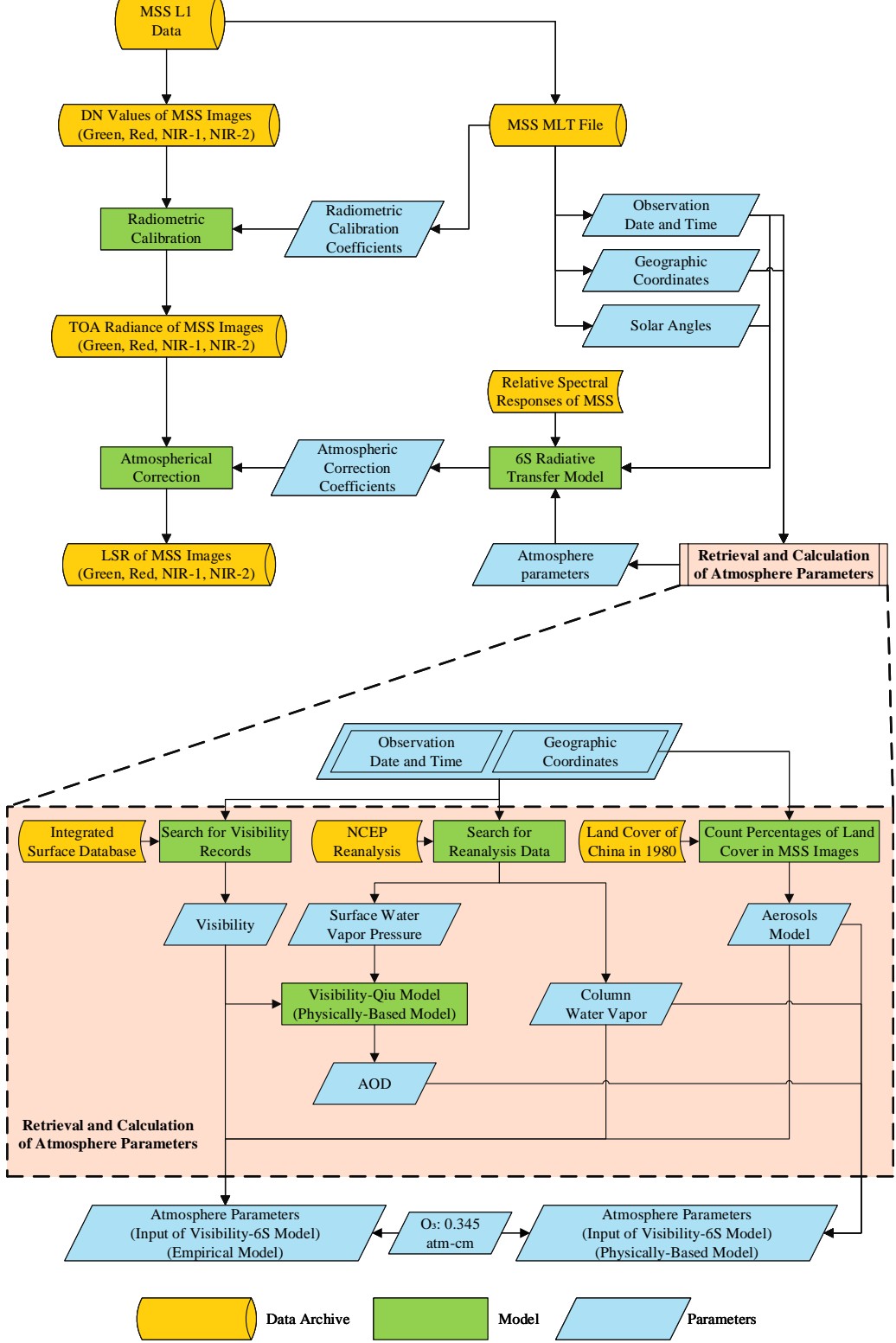

**Figure 3.** Workflow for generating the LSR data of MSS.

In the 6S model, an empirical model was proposed to describe the relationship between AOD and visibility. Two profiles of 23 km visibility and 5 km visibility were derived [42], and the AOD at 550 nm for visibility 5 km and 23 km were, respectively, calculated with the parameters defined [43–45]. For another visibility, a new particle density profile was computed by the profiles defined by the visibility of 5 km and 23 km.

### 3.3. AOD Retrieval Model Based on Visibility

In addition to the empirical models, several physically-based models were proposed on the concept of visibility proposed by Koschmieder [26]. Combined with the Elterman model describing the relationship between visibility and extinction coefficient of aerosols [27], a corrected scheme for China is proposed [29], as follows:

$$\tau_a = \left(\frac{3.912}{V} - 0.0116\right)\left(H_1\left(e^{-\frac{z}{H_1}} - e^{-\frac{5.5}{H_2}}\right) + 12.5e^{-\frac{5.5}{H_2}} + H_2 e^{\frac{-5.5}{H_1}}\right) \times f \tag{2}$$

where $\tau_a$ is the total optical depth of atmospheric molecules and aerosol particles, $V$ is the horizontal visibility, $H_1$ is 0.886 km + 0.0222 V, $H_2$ is 3.77 km, and $z$ is the height of visibility measurement. For northeastern China, the empirical correction coefficient $f = e^{-0.32+0.02}$ V. For other Chinese regions, the empirical correction coefficient $f = e^{(0.42+0.0046p_w+0.015V_z)}e^{\frac{-0.0047V^2}{p_w}}$, where $P_w$ is the water pressure of the visibility measurement location in hPa. The surface water pressure of stations is also estimated using NCEP/NCAR Reanalysis 1.

Though researchers proposed other advanced models to retrieve AOD from visibility, Qiu's model is more convenient and is easier to calculate, as all the input parameters can be calculated based on the NCEP/NCAR Reanalysis 1 and ISD. Therefore, we applied Qiu's AOD retrieval model in this research.

## 4. Evaluation and Uncertainty Analysis

Since rare simultaneous in situ LSR observation records were acquired in China about 30–50 years ago, we chose an indirect method to evaluate the MSS land surface reflectance product. Thanks to the simultaneous observation of MSS and TM in Landsat 4–5, the TM LEDAPS product was used as the truth value. Uncertainties are brought by several sources in the cross-validation, shown in Table 2 in detail when comparing the measured value with the truth value.

**Table 2.** The sources of uncertainty.

| Measured Value | Truth Value | RSR Difference | Radiometric Calibration | Geographical Factors | AOD/WV Estimating | Other Factors | Reference |
|---|---|---|---|---|---|---|---|
| LSR of TM (Visibility-6S and Visibility-Qiu Method) | LSR of TM (LEDAPS) | × | × | × | ○ | × | - |
| LSR of MSS (Visibility-6S and Visibility-Qiu Method) | LSR of TM (LEDAPS) | ○ | ○ | ○ | ○ | ○ | - |
| LSR of MSS (Visibility-6S and Visibility-Qiu Method) | LSR of TM (only pure pixel) (LEDAPS) | ○ | ○ | × | ○ | ○ | - |
| effective radiance or reflectance of MSS in TOA | effective radiance or reflectance of TM in TOA | ○ | × | × | × | × | Teixeira Pinto et al. [6] |
| radiance or reflectance of MSS in TOA | radiance or reflectance of TM in TOA | ○ | ○ | ○ | × | ○ | Teixeira Pinto et al. [6] |

○ means that the factor influences the validation results. × means that the factor does not influence the validation results.

### 4.1. Comparison Method

Three statistical metrics are used to evaluate the Landsat surface reflectance product, namely, accuracy, precision, and uncertainty (APU), as follows:

$$A = \frac{1}{n} \sum_{i=1}^{n} \varepsilon_i \qquad (3)$$

$$P = \sqrt{\frac{1}{n-1} \sum_{i=1}^{n} (\varepsilon_i - A)^2} \qquad (4)$$

$$U = \sqrt{\frac{1}{n} \sum_{i=1}^{n} \varepsilon_i^2} \qquad (5)$$

where the $\varepsilon_i = \rho_i - \rho_{i,reference}$ is the absolute error, and $\rho_{i,reference}$ is the LEDAPS surface reflectance, as the truth value. The global surface reflectance products of Landsat TM/ETM+ and OLI generated by LEDAPS and LaSRC are evaluated by these metrics [3,4]. Using these metrics can make the evaluation of the generated Landsat MSS LSR data consistent and comparable with the widely-used Landsat LSR products.

### 4.2. Uncertainty Brought by RSR

The differences between RSRs of TM and MSS may result in the least uncertainty when comparing the MSS LSR and TM LSR. The effective reflectance over MSS and TM spectral range can be calculated as follows:

$$\rho_B = \frac{\int \rho_H(\lambda) \times RSR(\lambda) d\lambda}{\int \rho_H(\lambda) d\lambda} \qquad (6)$$

where $\rho_H$ is the hyperspectral reflectance measured or simulated, while $\rho_B$ is the broadband reflectance, and the RSR is the relative spectral response of a specific band. By the broadband reflectance synthesized by hyperspectral reflectance, only the difference between RSRs is considered. Considering the land cover type, we assign the ECOSTRESS spectral library (v1.0) and PROSAIL simulated profiles to $\rho_H$.

The ECOSTRESS spectral library contains over 3000 spectra, including minerals, rocks, artificial materials, and vegetation, derived from the Advanced Spaceborne Thermal Emission Reflection Radiometer (ASTER) spectral library [46]. A part of the ECOSTRESS spectra provides records in the visible, near-infrared, and shortwave infrared bands, ranging from 0.35 μm to 2.4 μm. However, the ECOSTRESS spectral library only contains leaf spectral profiles, while the remote sensing sensors observe the vegetation canopy from the satellites. Therefore, we used the PROSAIL model [47] to simulate canopy spectra under various conditions. Table 3 shows the inputs of the PROSAIL model, generating 100,000 profiles.

Figures 4 and 5 show the uncertainty of the comparison, using TM effective reflectance as the truth value. Both the ECOSTRESS and PROSAIL datasets show a high level of bad evaluation with the largest uncertainty in the MSS NIR-1 band. The RSR shows the least correspondence (shown in Figure 1). The uncertainty ranges from 0.0062 (MSS red band) to 0.0472 (MSS NIR-1 band) in the ECOSTRESS dataset and ranges from 0.0056 (MSS Red Band) to 0.0986 (MSS NIR-1 band) in the PROSAIL dataset.

**Table 3.** Inputs of the PROSAIL model.

| Parameters | Value |
|---|---|
| Solar zenith angle | uniform (0, 70) |
| Observer zenith angle | 0 |
| Relative azimuth angle | uniform (0, 360) |
| Leaf area index | uniform (0, 8) |
| Equivalent water thickiness | uniform (0, 8) |
| Chlorophyll a + b concentration | uniform (10, 80) |
| Carotenoid concentration | uniform (0, 20) |
| Dry matter content | uniform (0.002, 0.010) |
| Brown pigment | 0 |
| PSOIL | uniform (0, 1) |
| typelidf | 2 |
| LIDFa | uniform (0, 90) |
| N | uniform (0.8, 2.5) |

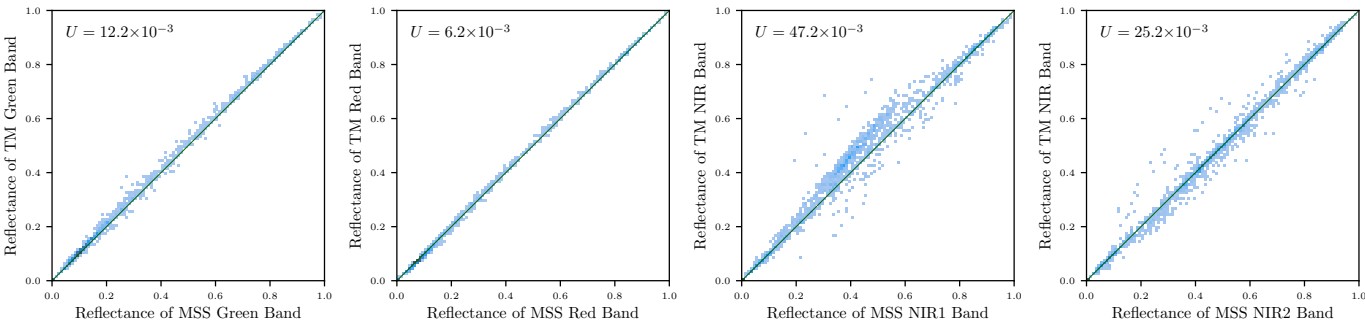

**Figure 4.** ECOSTRESS effective reflectance comparison of Landsat 5 MSS and TM.

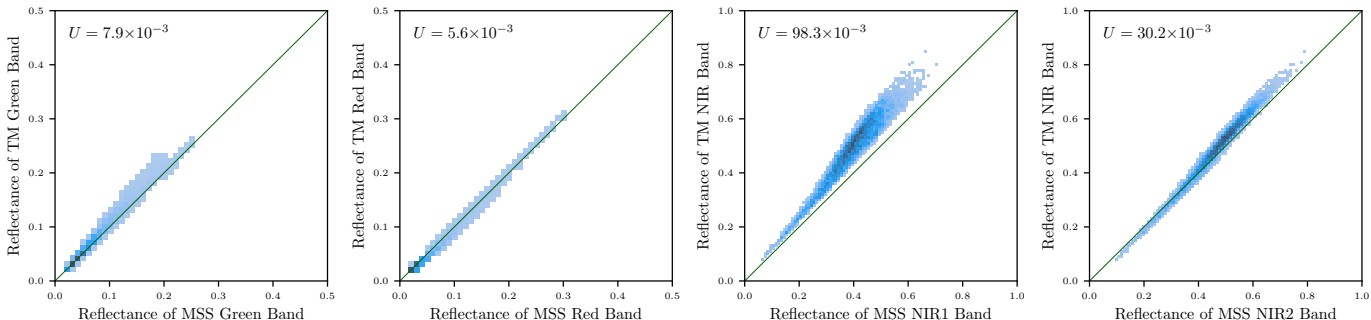

**Figure 5.** PROSAIL effective reflectance comparison of Landsat 5 MSS and TM.

The effective reflectance calculated by RSRs and spectral dataset performs much better than the actual image data (such as in the later Landsat 8 and Sentinel 2 remote sensing image pairs [48]) because actual image pairs are associated with many more factors influencing the uncertainty. As the endmember and abundance vary in actual remote sensing pixels, the uncertainty brought by RSRs may differ from the simulation. In general, the effective reflectance of TM and MSS shows good consistency in the red band but the worst consistency in the NIR band.

### 4.3. Uncertainty Brought by Georegistration and Scale Effects

When comparing MSS scenes and TM scenes, the error of geographical factors must be considered. A common practice is to resample the higher resolution image to the lower resolution image [48–50], if there are resolution inconsistencies for the image pairs. This step of processing is usually helpful but under the condition of higher georegistration error.

The following procedure extracts the pixel values of a pair of MSS and TM scenes (shown in Figure 6). First, generate random points within the range of MSS and TM scenes. The number of random points generated is 100,000 per image. Then, extract the pixel value of the MSS scene and TM scene according to the point, respectively. Finally, calculate the variance of the receptive field (near the TM pixel extracted) and set the upper limit of the variance. The receptive field is an $n \times n$ pixels window, with $n$ assigned to 1, 3, 5, 7, 9, and 11 and the upper limitation of the variance $v_{upperbound}$ is set as 0.002, 0.004, 0.006, 0.008, 0.01, 0.02, 0.03, 0.04, 0.05, 0.06, 0.07, 0.08, 0.09, and 0.10. Only the image in which the final number of random points is over 1000 is applied for the evaluation. In addition to A, P, and U, the coefficient of determination is used to evaluate the correlation of the MSS and TM pixels extracted, as follows:

$$R^2(\rho_{MSS}, \rho_{TM}) = 1 - \frac{\sum(\rho_{TM} - \rho_{MSS})^2}{\sum(\rho_{TM} - \overline{\rho_{TM}})^2} \tag{7}$$

where $\rho_{TM}$ and $\rho_{MSS}$ are the reflectance extracted by the random point, while $\overline{\rho_{TM}}$ is the average of $\rho_{TM}$.

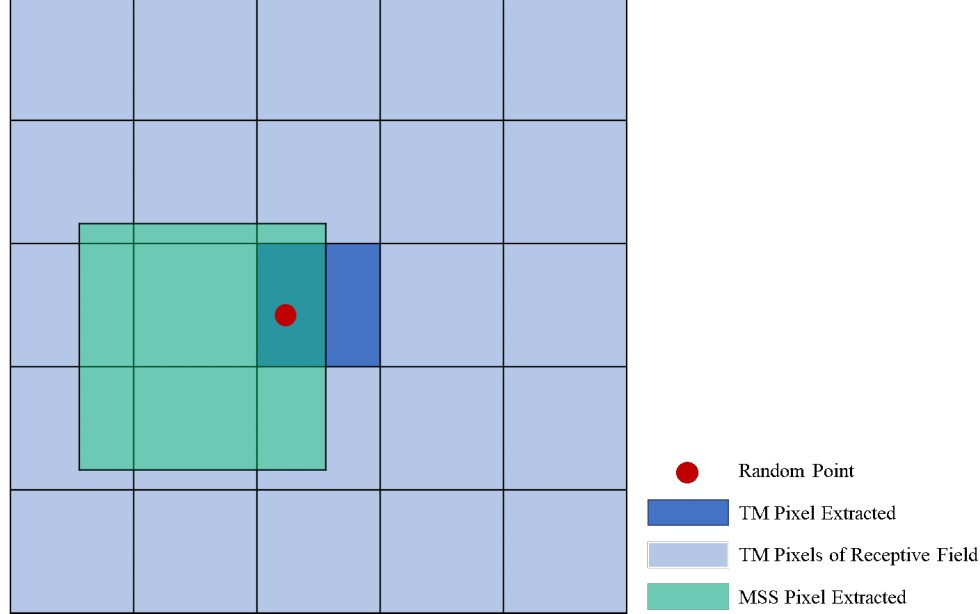

**Figure 6.** An example of MSS and TM pixels extract by random point.

Figures 7 and 8 show the results. By controlling the receptive field area and the upper limits of variance, we can quantitatively estimate the error of geographical factors and find the optimal receptive field and upper limitation of variance. For a certain upper limit $v_{upperbound}$, the larger receptive field causes the lower uncertainty. However, the maximum of $R^2$ is not reached with the minimum $v_{upperbound}$. Once the filter regulation of n and $v_{upperbound}$ is too strict, the number of random points used could become much smaller. A large-scale and homogeneous region is always associated with a minority of land cover, resulting in a localized distribution of the random point in the scatter diagram. Therefore, we regard $n = 3$ and $v_{upperbound} = 0.01$ as the proper value because the maximum of $R^2$ is always reached, which is a compromise of screening regulations and screening results.

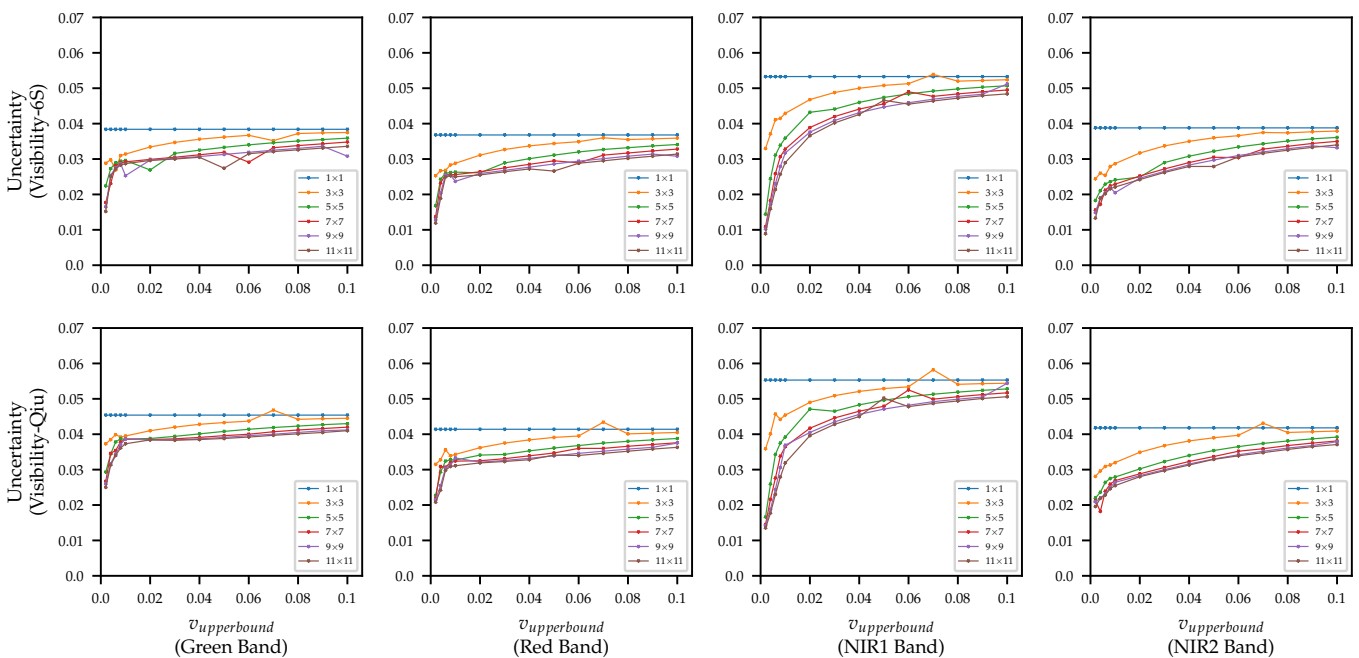

**Figure 7.** Relationship between $v_{\text{upper bound}}$ and uncertainty with different receptive field.

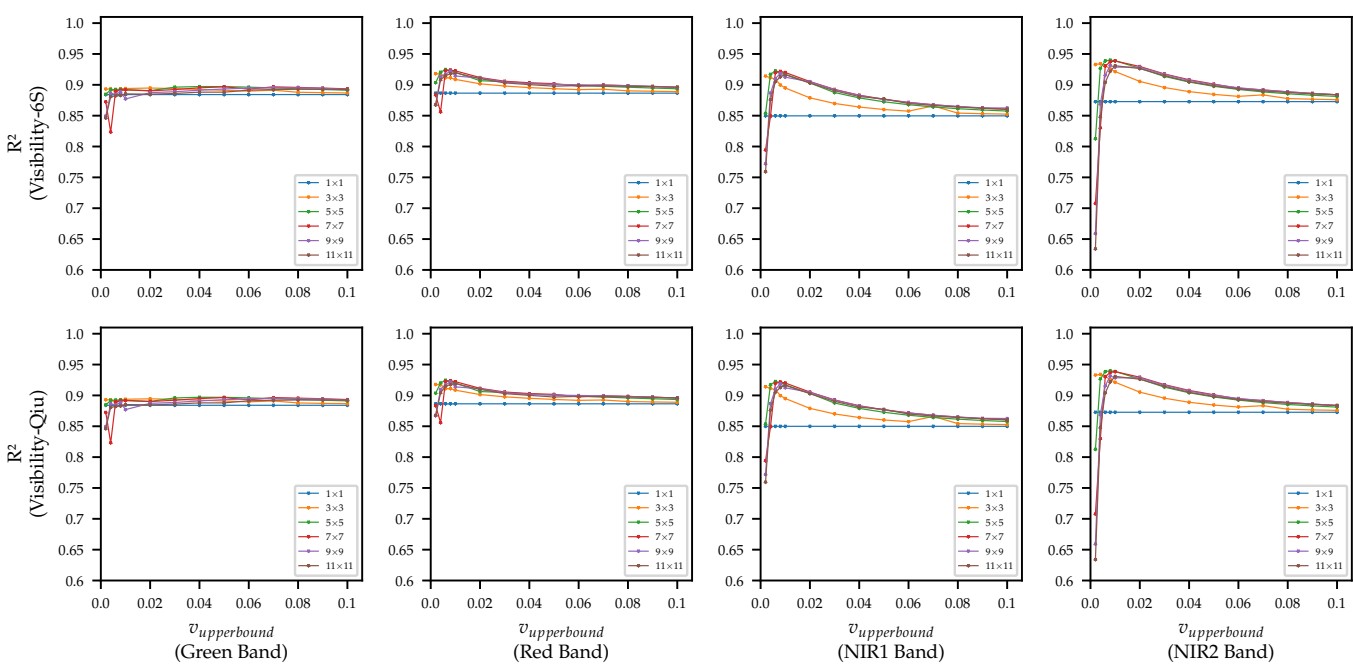

**Figure 8.** Relationship between $v_{\text{upper bound}}$ and $R^2$ with different receptive field.

### 4.4. Uncertainty Brought by AOD Estimation

By the filter regulation proposed, the final A, P, and U are calculated and are shown in Table 4, for the total 479 pairs with $n = 3$ and $v_{upperbound} = 0.01$, and the evaluation of TM LEDAPS products [3]. Not only are the MSS LSR data evaluated, but the TM LSR produced by the visibility–6S method and visibility–Qiu method are evaluated as well, because the validation of LSR produced by the visibility–6S method and visibility–Qiu using TM LEDAPS as the truth value can directly figure the uncertainty brought by different methods of AOD estimation, without any error from RSR difference, radiometric calibration, and

geographical factors. As shown in Table 4, the best performance of the visibility method is acquired by the visibility–6S method, with the uncertainty of 0.020 (TM green band), 0.014 (TM red band), 0.009 (TM NIR band), and the uncertainty of 0.029 (MSS green band), 0.026 (MSS red band), 0.033 (MSS NIR-1 band), and 0.023 (MSS NIR-2 band). The uncertainty also decreases from the green band to the NIR band for both visibility methods, except the MSS NIR-1 band. As A increases from the green band to the TM NIR band, an overestimation of AOD by visibility method may be systemic bias.

**Table 4.** APU scores of surface reflectance by visibility–6S method and visibility–Qiu method (compared with TM surface reflectance APU [3]).

| Sensor | Band | N | Comparison Metrics | | | | | | | | | | | |
| | | | Visibility–6S Compared with LEDAPS (This Work) | | | Visibility–Qiu Compared with LEDAPS (This Work) | | | TM LEDAPS Compared with MODIS BRDF [3] | | | LEDAPS Compared with LSR Using AERONET AOD [3] | | |
| TM | Green | 479 | −0.019 | 0.005 | 0.020 | −0.025 | 0.008 | 0.029 | 0.001 | 0.009 | 0.009 | 0.0001 | 0.0054 | 0.0054 |
| | Red | 479 | −0.012 | 0.004 | 0.014 | −0.016 | 0.007 | 0.021 | 0.009 | 0.01 | 0.014 | 0.0001 | 0.0041 | 0.0041 |
| | NIR | 479 | 0.003 | 0.005 | 0.009 | 0.005 | 0.008 | 0.016 | 0.005 | 0.017 | 0.017 | 0.0032 | 0.0061 | 0.0068 |
| MSS | Green | 479 | −0.023 | 0.013 | 0.029 | −0.029 | 0.016 | 0.039 | - | - | - | - | - | - |
| | Red | 479 | −0.019 | 0.015 | 0.026 | −0.023 | 0.017 | 0.032 | - | - | - | - | - | - |
| | NIR1 | 479 | −0.017 | 0.020 | 0.033 | −0.017 | 0.022 | 0.037 | - | - | - | - | - | - |
| | NIR2 | 479 | 0.008 | 0.015 | 0.023 | 0.010 | 0.016 | 0.027 | - | - | - | - | - | - |

As the precision promotion of atmospheric correction could not directly reduce the uncertainty brought by the other factors, we divide the uncertainty sources into atmospheric and non-atmospheric factors. The uncertainty of non-atmospheric factors could be computed with the following two assumptions. One is that the atmospheric factors and non-atmospheric factors are independent. The other is that the MSS uncertainty brought by the atmosphere equals the TM uncertainty brought by the atmosphere for the similar bands. The results (shown in Table 5) show that the uncertainty brought by atmospheric factors is smaller than the uncertainty brought by non-atmospheric factors during the cross-comparison, which also suggests that the non-atmospheric factors cannot be ignored when using the MSS LSR data for time-series analysis.

**Table 5.** Uncertainty sources of surface reflectance by visibility–6S method and visibility–Qiu method.

| Band | MSS Visibility–6S Method | | | MSS Visibility–Qiu Method | | |
| | Total Uncertainty | Uncertainty Brought by Atmospheric Factors | Uncertainty Brought by Non-Atmospheric Factors | Total Uncertainty | Uncertainty Brought by Atmospheric Factors | Uncertainty Brought by Non-Atmospheric Factors |
| Green | 0.029 | 0.020 | 0.021 | 0.039 | 0.029 | 0.025 |
| Red | 0.026 | 0.014 | 0.022 | 0.032 | 0.021 | 0.025 |
| NIR1 | 0.033 | 0.009 | 0.032 | 0.037 | 0.016 | 0.033 |
| NIR2 | 0.023 | 0.009 | 0.021 | 0.027 | 0.016 | 0.022 |

*4.5. Other Uncertainty Sources*

Moreover, it is necessary to note that the uncertainty of the inter-comparison is a combination, including the uncertainty of the truth value. As reported, the dense dark vegetation method for AOD retrieval may not be suitable for the clamps in Australia [51,52] or for other positions where the vegetation is not dense and dark enough [53]. Table 6 shows the APU of the four seasons, respectively. The performance of the visibility method of winter (December, January, and February, DJF) is much worse than the other seasons. The reason might be explained by several aspects: (1) The dense dark vegetation pixels are insufficient, as the major vegetation is broad-leaved trees in northeast and North China. The surface reflectance of LEDAPS treated as the truth value may have higher uncertainty; (2) The solar elevation angle in winter is higher than in other seasons, causing a longer optical path and increasing the uncertainty brought by AOD estimation.

Additionally, occasional anomalies may cause an increase in uncertainty, especially in the earlier sensors. Figure 9 shows an example of oversaturation and detector striping

in MSS images. Occasional anomalies occur more frequently in the early sensors and affect the final validation results, and the influence needs to be estimated quantitatively for further research.

**Table 6.** APU scores by seasons of LSR by visibility–6S method and visibility–Qiu method (MAM: March, April, May; JJA: June, July, August; SON: September, October, November; DJF: December, January, February).

| Band | Season | N | Comparison metrics | | | | | | | | | | | |
|------|--------|---|---|---|---|---|---|---|---|---|---|---|---|---|
| | | | Visibility–6S TM | | | Visibility–Qiu TM | | | Visibility–6S MSS | | | Visibility–Qiu MSS | | |
| | | | **A** | **P** | **U** | **A** | **P** | **U** | **A** | **P** | **U** | **A** | **P** | **U** |
| Green | MAM | 167 | −0.017 | 0.003 | 0.017 | −0.020 | 0.004 | 0.022 | −0.025 | 0.013 | 0.030 | −0.028 | 0.014 | 0.035 |
| | JJA | 113 | −0.013 | 0.002 | 0.014 | −0.021 | 0.003 | 0.021 | −0.019 | 0.009 | 0.022 | −0.027 | 0.009 | 0.029 |
| | SON | 123 | −0.022 | 0.004 | 0.023 | −0.035 | 0.009 | 0.038 | −0.024 | 0.011 | 0.029 | −0.037 | 0.015 | 0.044 |
| | DJF | 76 | −0.026 | 0.012 | 0.032 | −0.026 | 0.021 | 0.042 | −0.023 | 0.022 | 0.040 | −0.022 | 0.031 | 0.052 |
| Red | MAM | 167 | −0.012 | 0.002 | 0.012 | −0.014 | 0.003 | 0.016 | −0.023 | 0.017 | 0.029 | −0.025 | 0.017 | 0.032 |
| | JJA | 113 | −0.010 | 0.002 | 0.010 | −0.016 | 0.003 | 0.017 | −0.014 | 0.015 | 0.022 | −0.021 | 0.015 | 0.027 |
| | SON | 123 | −0.014 | 0.004 | 0.015 | −0.022 | 0.008 | 0.026 | −0.017 | 0.010 | 0.021 | −0.026 | 0.014 | 0.032 |
| | DJF | 76 | −0.013 | 0.010 | 0.021 | −0.011 | 0.018 | 0.032 | −0.020 | 0.018 | 0.031 | −0.018 | 0.026 | 0.041 |
| NIR1 | MAM | 167 | 0.002 | 0.004 | 0.007 | 0.003 | 0.005 | 0.010 | −0.016 | 0.018 | 0.028 | −0.017 | 0.018 | 0.029 |
| | JJA | 113 | 0.000 | 0.004 | 0.007 | 0.000 | 0.005 | 0.012 | −0.032 | 0.023 | 0.043 | −0.034 | 0.022 | 0.043 |
| | SON | 123 | 0.002 | 0.005 | 0.008 | 0.003 | 0.010 | 0.017 | −0.017 | 0.018 | 0.029 | −0.018 | 0.020 | 0.035 |
| | DJF | 76 | 0.010 | 0.012 | 0.019 | 0.018 | 0.019 | 0.032 | 0.005 | 0.023 | 0.034 | 0.012 | 0.030 | 0.046 |
| NIR2 | MAM | 167 | - | - | - | - | - | - | 0.008 | 0.013 | 0.021 | 0.008 | 0.014 | 0.022 |
| | JJA | 113 | - | - | - | - | - | - | 0.000 | 0.014 | 0.020 | −0.001 | 0.014 | 0.022 |
| | SON | 123 | - | - | - | - | - | - | 0.012 | 0.015 | 0.026 | 0.013 | 0.018 | 0.032 |
| | DJF | 76 | - | - | - | - | - | - | 0.017 | 0.018 | 0.028 | 0.025 | 0.021 | 0.037 |

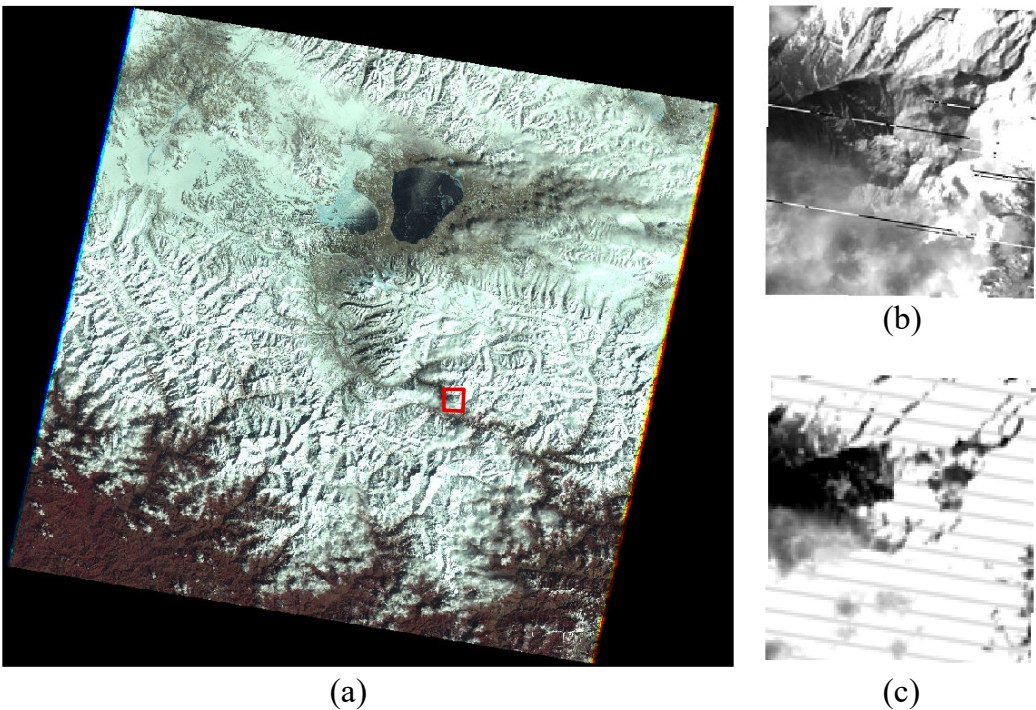

**Figure 9.** An example of occasional anomalies (occurred in MSS and TM images of 144039_19890222). (**a**) False-color synthesis of Landsat 5 TM RGB composite (red: B4, green: B3, blue: B2). (**b**) An example of detector failure of Landsat 5 TM Band 4 (NIR band). (**c**) An example of detector striping and oversaturation of Landsat 5 MSS and 3 (NIR-1 band).

## 5. Application of Time-Series SR in Spectral-Stable Land Cover

To examine the stability of the visibility-based surface reflectance of MSS in practice, we selected five ROIs across China, extracted the values of MSS, TM, and MODIS reflectance of the ROIs, and conducted time-series assessments. The ROIs cover three kinds of land cover, i.e., water body (ROI 1), desert (ROI 2 and 5), and evergreen vegetation (ROI 3 and 4), shown in Figure 10.

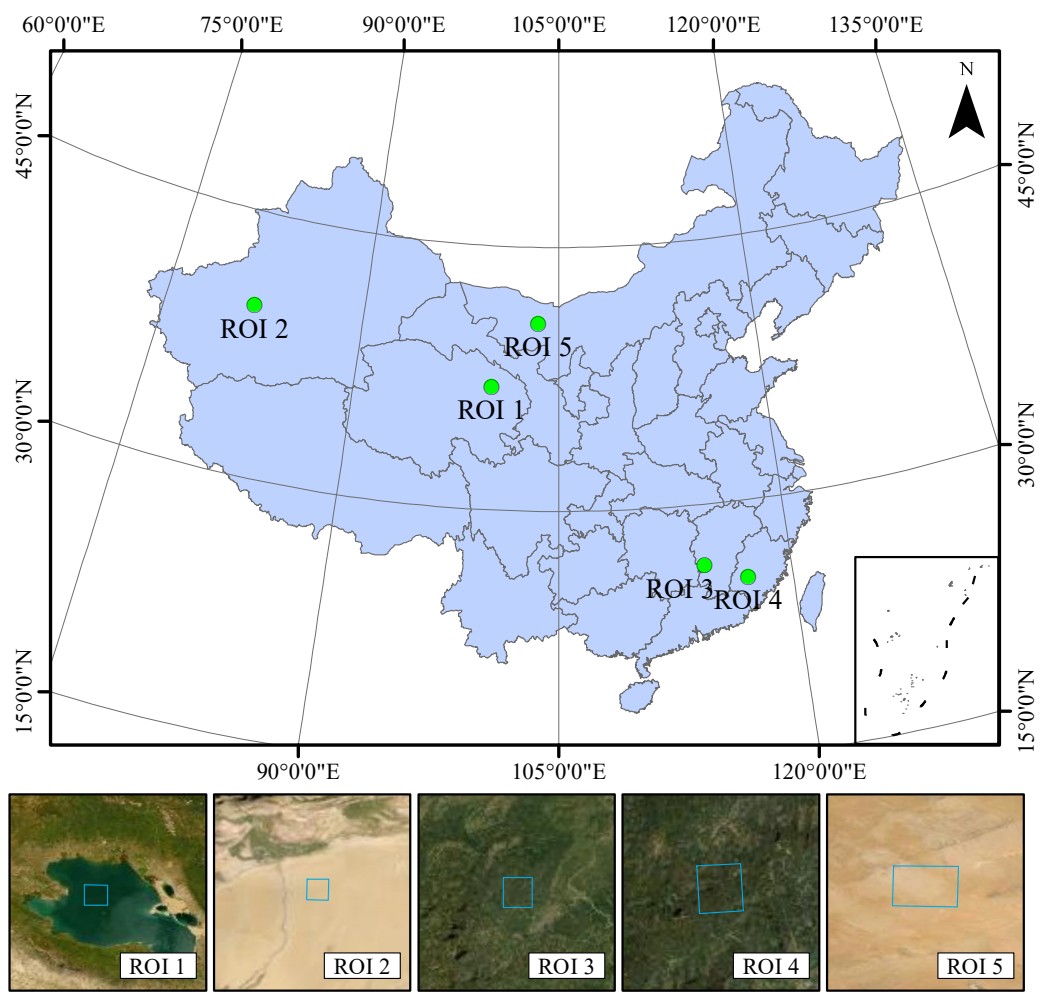

**Figure 10.** ROIs of spectral-stable land cover in China.

The MSS LSR data are the atmospherically corrected image based on the visibility–6S method, which performs better than the visibility–Qiu method in the validation mentioned above. The TM LSR data are from the LEDAPS data, and the MODIS data used are from the MOD09A1.006 product, an eight-day composition with a 500 m resolution. The ROIs are selected by the time-series reflectance by the MODIS during the 20 years using the Google Earth Engine. Considering the resolution inconsistency of MODIS, TM, and MSS, though the ROIs are selected for spatial homogeneity in MODIS images, they are always heterogeneous in the smaller pixels of TM and MSS images. Here, we used the median value of ROIs of all three LSR data for times-series assessment, which can also remove many outliers caused by MSS defective sensor. All the remote sensing data used are free from cloud, with strict control by the cloud coverage and artificial visual interpretation.

Though the surface reflectance cannot be less than zero in practical terms, the value of remote sensing LSR product may sometimes be less than zero in some dark pixels, for the overestimation of AOD and WV causes the overestimation of path radiance. In addition, the shadow of cloud and topographic relief may cause the LSR to be less than zero. The

negative values are found in all three LSR data used, without artificial interventions in later assessments.

### 5.1. Water Body

ROI 1 is an area of water body, Qinghai Lake, in Qinghai Province, between 36°55′–37°3′N, 100°0′–100°10′. Qinghai Lake is the largest saline lake in China and is also one of the remote sensing test fields of China, lying on the northeastern Tibetan Plateau at the average altitude of 3200 m [54]. As Qinghai Lake freezes in winter, all the remote sensing images used for ROI 1 are acquired during the unfreezing stage.

Figure 11 shows the time series of the MSS LSR, TM-LEDAPS, and MODIS-MOD09A1 (simultaneous with the TM image). Among all the MSS data exhibited in Figure 11, outliers are found in the image of LM02_L1GS_143034_19750621_20180425_01_T2, especially for the green band. The TOA reflectance of ROI 1 calculated by the rescaling coefficients provided by the MTL file is less than zero. It may be caused by inappropriate radiometric calibration coefficients or invalid DN values derived from unknown sensor issues, which needs further research.

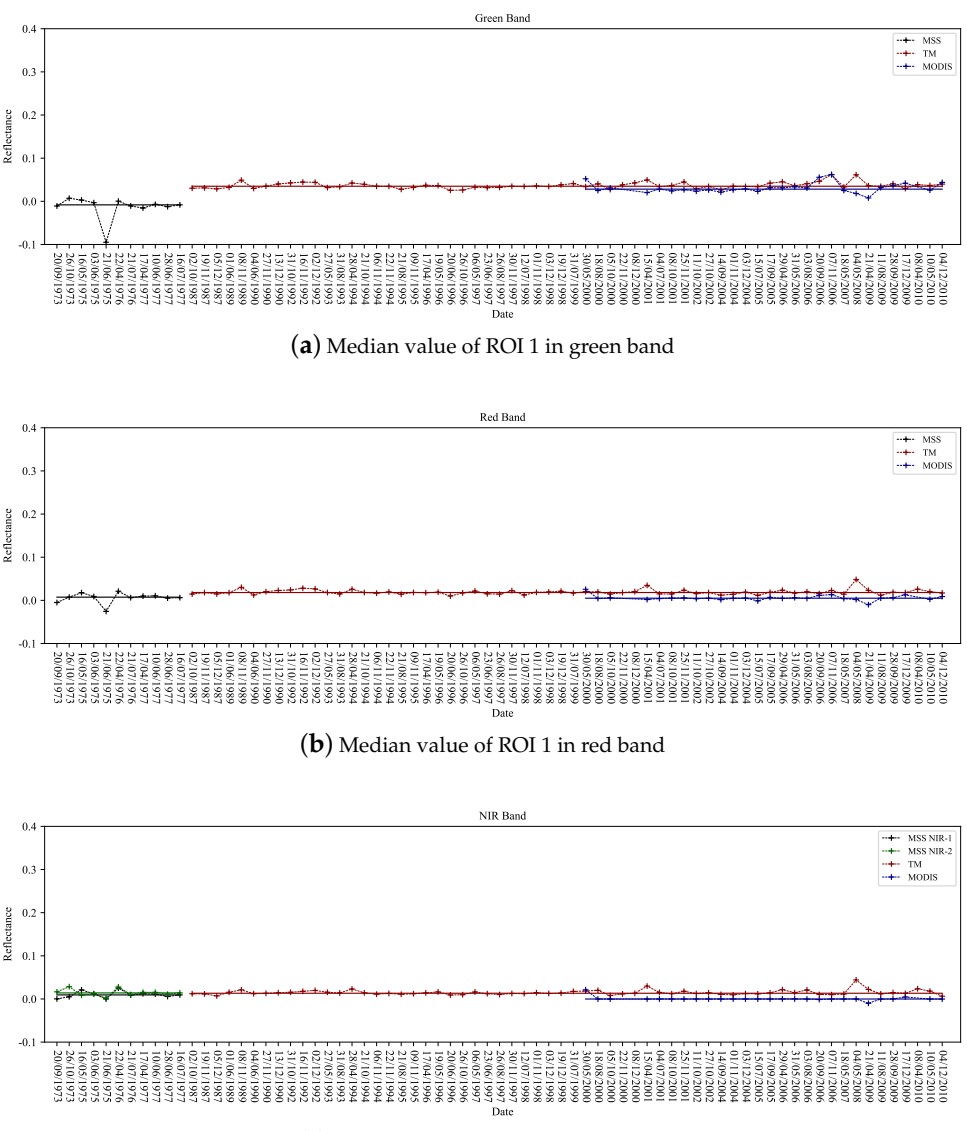

(**a**) Median value of ROI 1 in green band

(**b**) Median value of ROI 1 in red band

(**c**) Median value of ROI 1 in NIR band

**Figure 11.** Time series of ROI 1 median value (ROI 1: water body).

Removing the outliers mentioned above, Table 7 shows the statistics of time-series reflectance of ROI-1 of the three sensors, including the minimum, median, mean, maximum, and standard deviation (SD). The reflectance of the MSS green band ranges from −0.095 to 0.007, which is smaller than the reflectance of green bands of TM (ranging from 0.026 to 0.062) and MODIS (ranging from 0.008 to 0.062). In red band, MSS reflectance ranges from −0.026 to 0.021, between TM reflectance (ranging from 0.010 to 0.048) and the MODIS reflectance (ranging from −0.010 to 0.026). The reflectance of MSS NIR bands ranges from 0.000 to 0.025 (MSS NIR-1 band) and from 0.002 to 0.029, which is similar to the TM NIR bands (ranging from 0.006 to 0.044) and is greater than the MODIS NIR band (ranging from −0.010 to 0.021).

**Table 7.** Statistics of time-series reflectance of MSS, TM, and MODIS in ROI 1 (ROI 1: water body).

| Band | Sensor | Min | Median | Mean | Max | SD |
|---|---|---|---|---|---|---|
| | MSS | −0.095 | −0.008 | −0.014 | 0.007 | 0.026 |
| Green | TM | 0.026 | 0.035 | 0.037 | 0.062 | 0.007 |
| | MODIS | 0.008 | 0.028 | 0.031 | 0.062 | 0.012 |
| | MSS | −0.026 | 0.007 | 0.006 | 0.021 | 0.012 |
| Red | TM | 0.010 | 0.018 | 0.019 | 0.048 | 0.006 |
| | MODIS | −0.010 | 0.005 | 0.006 | 0.026 | 0.006 |
| | MSS-NIR1 | 0.000 | 0.009 | 0.010 | 0.025 | 0.007 |
| | MSS-NIR2 | 0.002 | 0.014 | 0.015 | 0.029 | 0.008 |
| NIR | TM | 0.006 | 0.013 | 0.015 | 0.044 | 0.006 |
| | MODIS | −0.010 | 0.000 | 0.001 | 0.021 | 0.005 |

### 5.2. Desert

ROI 2 and ROI 5 are areas of desert, as stable high-reflectance regions. ROI 2 is located in northwestern Taklimakan Desert, in Xinjiang Province, between 39°26′–39°54′N, 81°24′–81°54′E. ROI 5 is located in the center of Badain Jaran Desert, in Inner Mongolia, between 40°33′–40°41′N, 103°18′–103°31′E.

Figures 12 and 13 show the time series reflectances of ROI 2 and ROI 5, which vary in a larger range than water body. Tables 8 and 9 show the detailed statistics of time-series reflectance of MSS, TM, and MODIS in ROI 2 and ROI 5. In ROI 2, the differences of each group of statistics are smaller than the validation result in Section 4, which shows a good time continuity. Nevertheless, in ROI 5, the reflectance of green bands shows an obvious difference, which is found in the comparison not only between MSS and TM reflectance but also between TM and MODIS reflectance. For red and NIR bands, the reflectance of MSS shows a better consistency compared with the green band.

**Table 8.** Statistics of time-series reflectance of MSS, TM, and MODIS in ROI 2 (ROI 2: desert).

| Band | Sensor | Min | Median | Mean | Max | SD |
|---|---|---|---|---|---|---|
| | MSS | 0.211 | 0.237 | 0.236 | 0.275 | 0.020 |
| Green | TM | 0.213 | 0.247 | 0.247 | 0.282 | 0.015 |
| | MODIS | 0.211 | 0.228 | 0.230 | 0.282 | 0.014 |
| | MSS | 0.271 | 0.298 | 0.296 | 0.312 | 0.012 |
| Red | TM | 0.262 | 0.295 | 0.294 | 0.322 | 0.012 |
| | MODIS | 0.265 | 0.287 | 0.289 | 0.345 | 0.016 |
| | MSS NIR-1 | 0.291 | 0.328 | 0.326 | 0.346 | 0.016 |
| | MSS NIR-2 | 0.289 | 0.339 | 0.333 | 0.362 | 0.018 |
| NIR | TM | 0.292 | 0.316 | 0.316 | 0.347 | 0.010 |
| | MODIS | 0.290 | 0.314 | 0.315 | 0.369 | 0.016 |

**Table 9.** Statistics of time-series reflectance of MSS, TM, and MODIS in ROI 5 (ROI 5: desert).

| Band | Sensor | Min | Median | Mean | Max | SD |
|------|--------|-----|--------|------|-----|-----|
| Green | MSS | 0.158 | 0.184 | 0.183 | 0.208 | 0.017 |
| | TM | 0.208 | 0.232 | 0.231 | 0.271 | 0.009 |
| | MODIS | 0.178 | 0.209 | 0.207 | 0.222 | 0.010 |
| Red | MSS | 0.252 | 0.283 | 0.281 | 0.315 | 0.017 |
| | TM | 0.274 | 0.298 | 0.298 | 0.333 | 0.010 |
| | MODIS | 0.248 | 0.293 | 0.290 | 0.312 | 0.014 |
| NIR | MSS NIR-1 | 0.302 | 0.344 | 0.339 | 0.373 | 0.019 |
| | MSS NIR-2 | 0.318 | 0.364 | 0.361 | 0.392 | 0.025 |
| | TM | 0.306 | 0.335 | 0.335 | 0.365 | 0.009 |
| | MODIS | 0.296 | 0.345 | 0.341 | 0.370 | 0.015 |

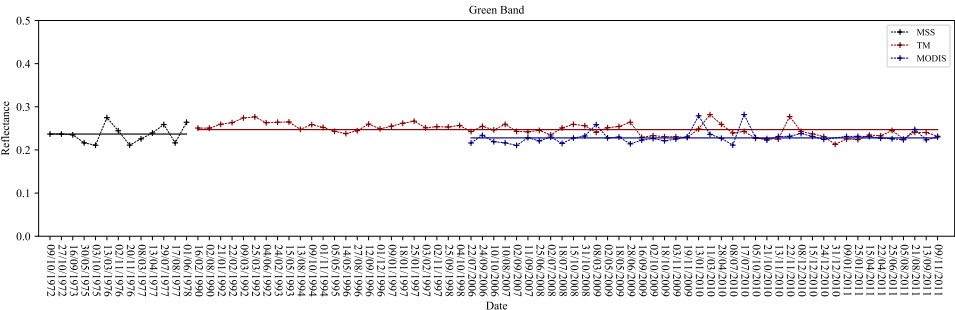

(**a**) Median value of ROI 2 in green band

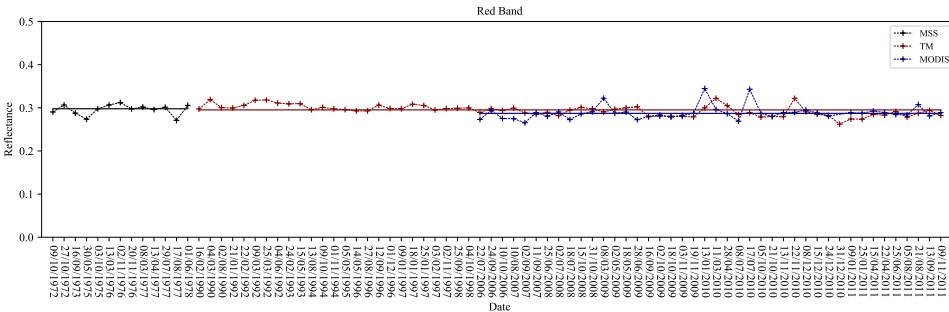

(**b**) Median value of ROI 2 in red band

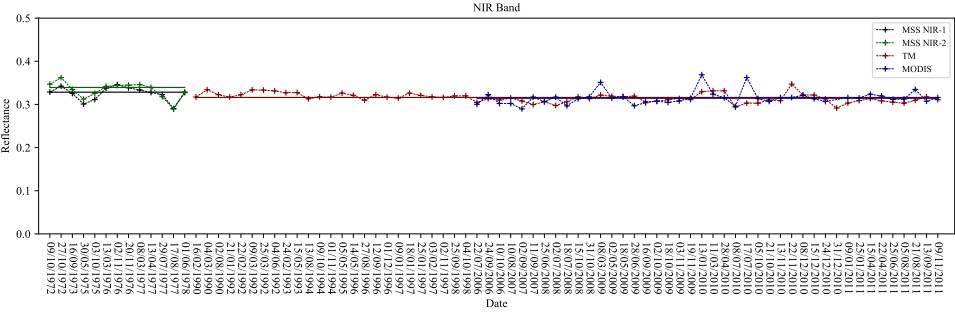

(**c**) Median value of ROI 2 in NIR band

**Figure 12.** Time series of ROI 2 median value (ROI 2: desert).

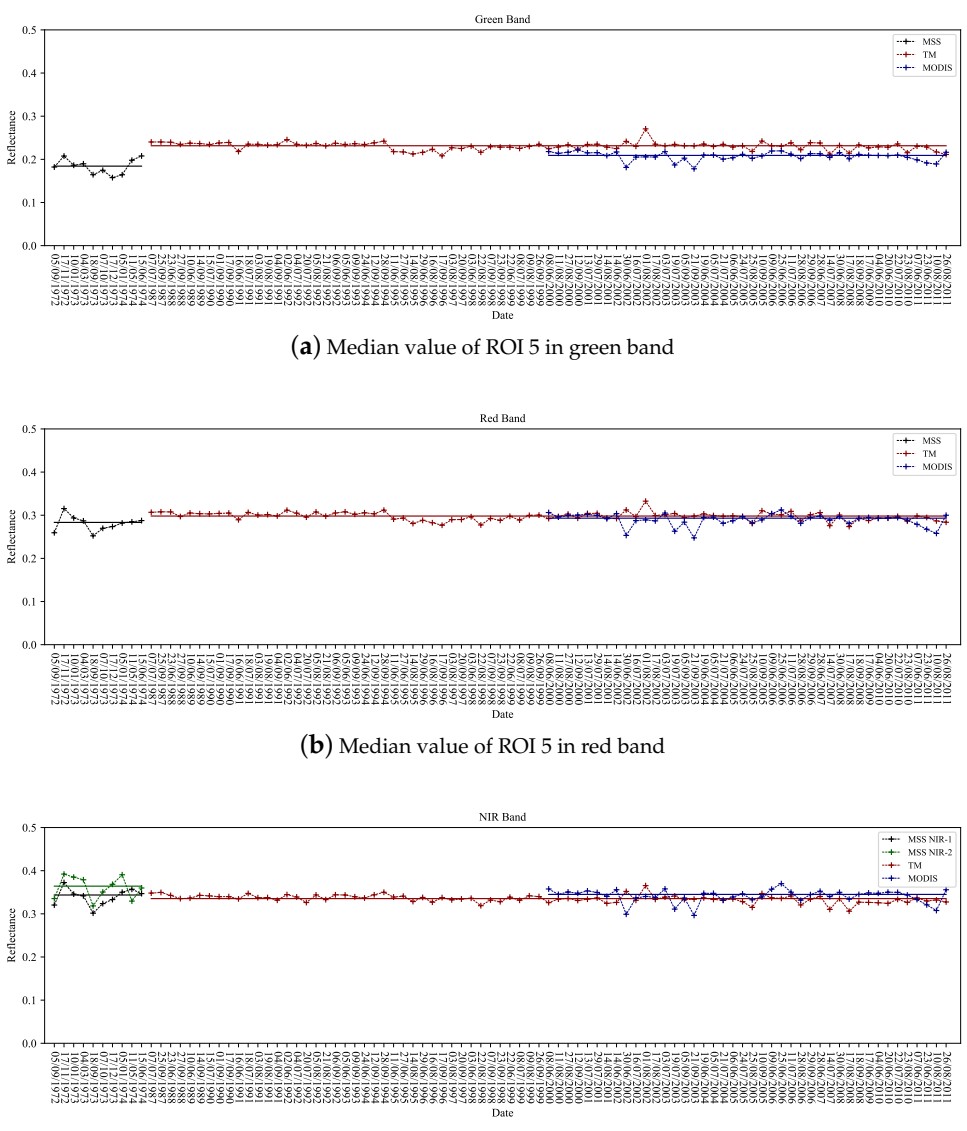

(**a**) Median value of ROI 5 in green band

(**b**) Median value of ROI 5 in red band

(**c**) Median value of ROI 5 in NIR band

**Figure 13.** Time series of ROI 5 median value (ROI 5: desert).

*5.3. Vegetation*

ROI 3 and ROI 4 are areas of vegetation. ROI 3 is located in Fujian Province, between 26°25′–26°35′N and 114°19′–114°29′E, and ROI 4 is located in Jiangxi Province, between 25°31′–25°39′N and 117°5′–117°12′E. The major vegetation of the two ROIs is subtropical evergreen broad-leaved forest. Furthermore, we select the months with small changes of vegetation spectrum according to the TM observation.

Figures 14 and 15 show the time series reflectance of ROI 3 and ROI 4. Times series reflectance of TM and MODIS shows stable lines in the green and red bands, while the reflectance of the NIR band shows a wide distribution over observation date. For the green band, MSS reflectance is less than TM and MODIS reflectance in both ROI 3 and 4, which is also found in the evaluation in Section 4. For the red band, the MSS reflectance performs better than the validation result summarized in the previous section. For the NIR bands, the performance of the MSS NIR-2 band is consistent with the NIR band of TM and MODIS, while the MSS NIR-1 band shows a good correlation but lower values compared with the MSS NIR-1 band, as the MSS NIR-1 band observes vegetation red-edge spectrum approximately. Tables 10 and 11 show the detailed statistics of time-series reflectance of MSS, TM, and MODIS, in ROI 3 and ROI 4. Both ROI 3 and 4 show good consistency in the

MSS red and NIR-2 band compared with later TM and MSS LSR data, for RSRs of these bands are more similar than the other bands. It suggests that the Normalized Difference Vegetation Index (NDVI) should be calculated by the MSS red and NIR-2 bands.

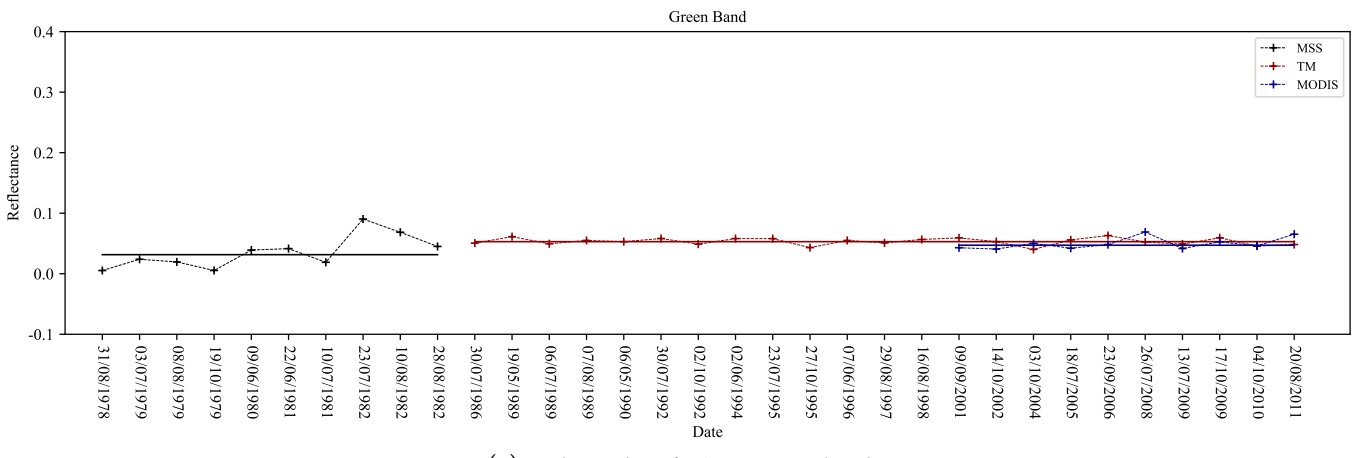

(**a**) Median value of ROI 3 in green band

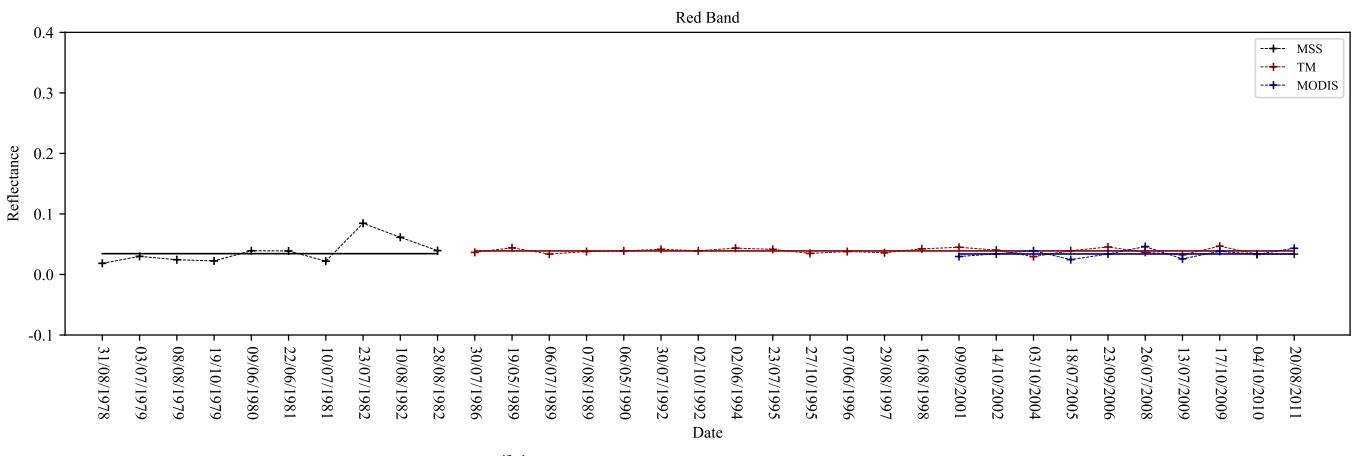

(**b**) Median value of ROI 3 in red band

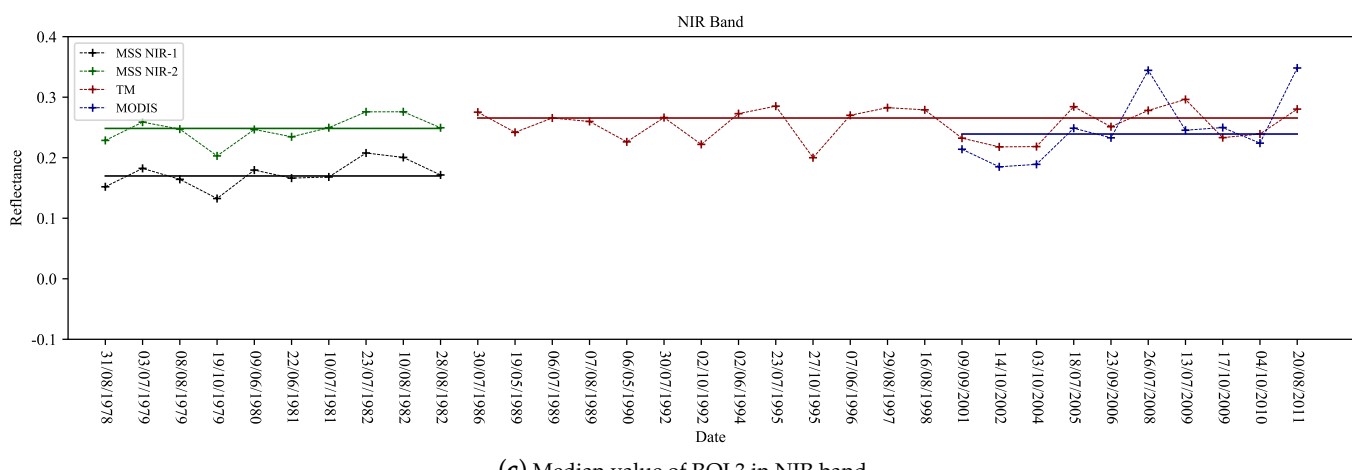

(**c**) Median value of ROI 3 in NIR band

**Figure 14.** Time series of ROI 3 median value (ROI 3: vegetation).

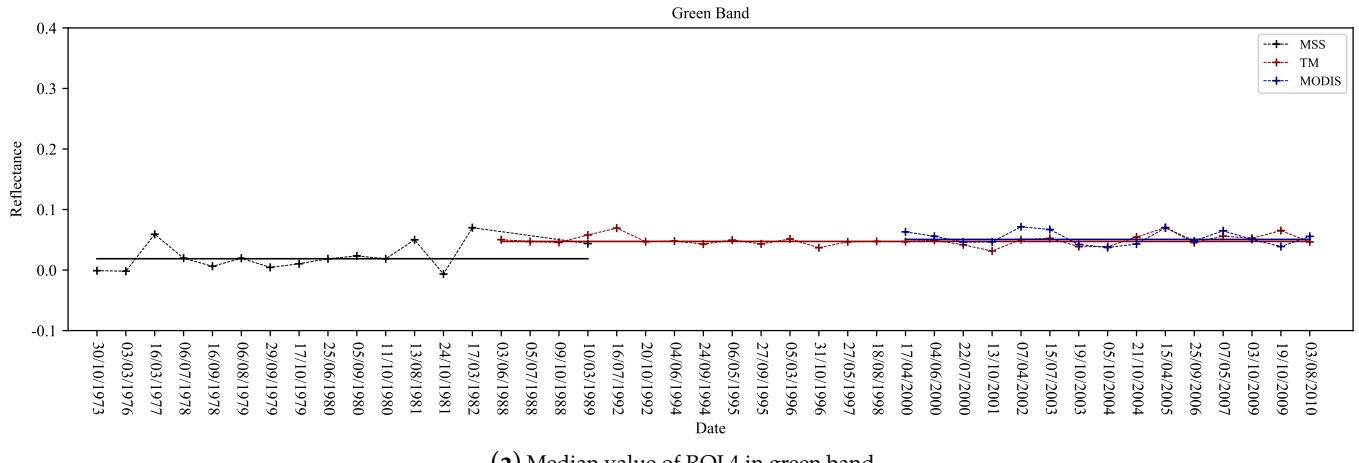

(**a**) Median value of ROI 4 in green band

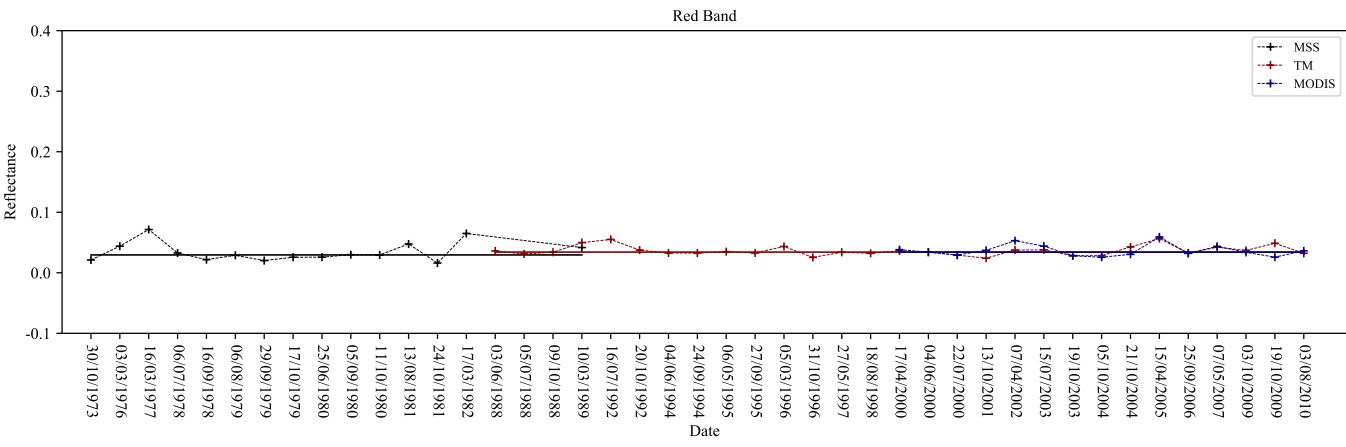

(**b**) Median value of ROI 4 in red band

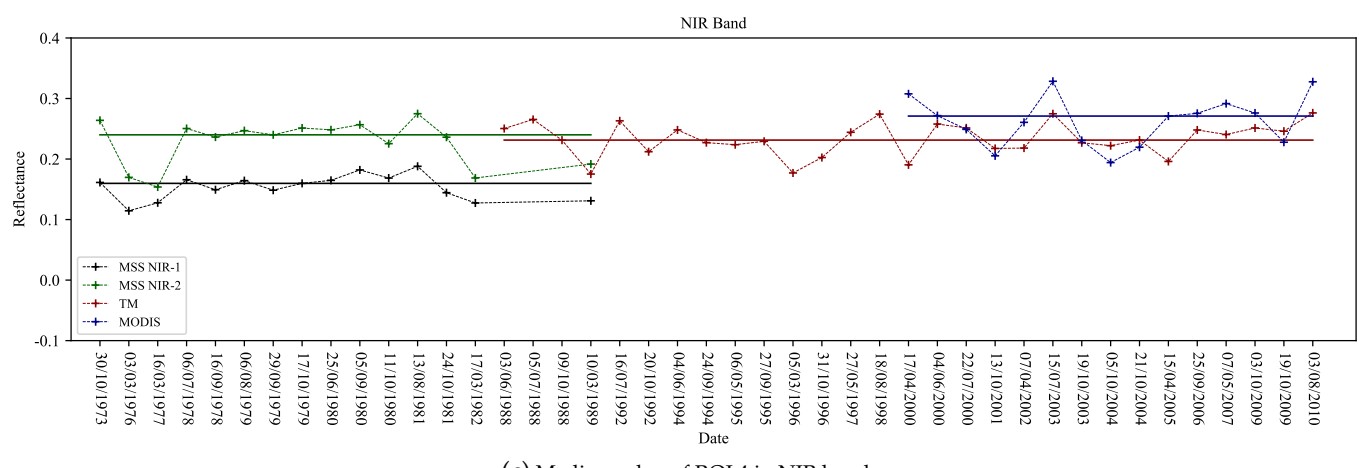

(**c**) Median value of ROI 4 in NIR band

**Figure 15.** Time series of ROI 4 median value (ROI 4: vegetation).

**Table 10.** Statistics of time-series reflectance of MSS, TM, and MODIS in ROI 3 (ROI 3: vegetation).

| Band | Sensor | Min | Median | Mean | Max | SD |
|---|---|---|---|---|---|---|
| Green | MSS | 0.005 | 0.032 | 0.036 | 0.090 | 0.026 |
| | TM | 0.040 | 0.053 | 0.053 | 0.063 | 0.006 |
| | MODIS | 0.041 | 0.047 | 0.050 | 0.069 | 0.009 |
| Red | MSS | 0.018 | 0.034 | 0.038 | 0.085 | 0.020 |
| | TM | 0.030 | 0.039 | 0.039 | 0.047 | 0.005 |
| | MODIS | 0.025 | 0.034 | 0.035 | 0.046 | 0.007 |
| NIR | MSS NIR-1 | 0.133 | 0.170 | 0.173 | 0.208 | 0.021 |
| | MSS NIR-2 | 0.203 | 0.248 | 0.247 | 0.276 | 0.021 |
| | TM | 0.200 | 0.266 | 0.256 | 0.296 | 0.027 |
| | MODIS | 0.185 | 0.239 | 0.248 | 0.348 | 0.054 |

**Table 11.** Statistics of time-series reflectance of MSS, TM, and MODIS in ROI 4 (ROI 4: vegetation).

| Band | Sensor | Min | Median | Mean | Max | SD |
|---|---|---|---|---|---|---|
| Green | MSS | −0.007 | 0.019 | 0.022 | 0.070 | 0.023 |
| | TM | 0.032 | 0.047 | 0.049 | 0.070 | 0.009 |
| | MODIS | 0.037 | 0.051 | 0.053 | 0.072 | 0.011 |
| Red | MSS | 0.016 | 0.030 | 0.035 | 0.072 | 0.016 |
| | TM | 0.024 | 0.034 | 0.037 | 0.057 | 0.008 |
| | MODIS | 0.026 | 0.034 | 0.037 | 0.059 | 0.009 |
| NIR | MSS NIR-1 | 0.115 | 0.160 | 0.153 | 0.188 | 0.020 |
| | MSS NIR-2 | 0.154 | 0.240 | 0.228 | 0.275 | 0.037 |
| | TM | 0.175 | 0.231 | 0.234 | 0.276 | 0.027 |
| | MODIS | 0.194 | 0.271 | 0.263 | 0.329 | 0.040 |

## 6. Discussion

### 6.1. Comparison of the Proposed Framework and LEDAPS

LEDAPS is a successful remote sensing image processing system that generates TM and ETM+ LSR products of Landsat 4, 5, and 7, based on the 6S model. Considering the lack of ozone data in the early years of remote sensing, the new framework proposed in this paper set the ozone data as a contrast value of 0.345 atm-cm. Topographic details within a scene of an MSS image are ignored due to our limited computing capability.

The major difference between the proposed framework and LEDAPS is that the ground-based visibility record is used as the input of 6S instead of the image-based retrieval of AOD. The benefit is that the image-based AOD retrieval method of MSS images is not needed. The image-based AOD retrieval method of MSS images is difficult to study today due to the limitation of AOD measurement in the 1970s and the inherent defects of MSS images. In addition, the precision of the image-based AOD retrieval method is influenced by the reported issues of image quality.

Moreover, LEDAPS uses the QA band as input in the process of generating SR to obtain the cloud mask [55], but the cloud mask of MSS images' QA band seems problematic [56]. The ground-based visibility records are independent of the remote sensing observation, so our proposed framework does not need the cloud mask.

As shown in Section 4, the results generated by our proposed framework have a larger uncertainty with a systemic bias compared to the LEDAPS LSR product of Landsat TM. The visibility-based AOD methods are regional, while artificial observations and weather conditions are also reported as error sources [57]. In our proposed framework, using a visibility record to generate MSS LSR data compromises accuracy and solvability. Although the proposed framework has a larger uncertainty than the LEDAPS, the systemic bias can be solved by the accumulating records of ground-based visibility and AOD results in recent decades.

*6.2. Analysis of Potential Problems in Time-Series Analysis of Early Years*

In addition to the uncertainty brought by atmospheric correction, a much larger uncertainty was found when directly comparing the MSS LSR generated by our proposed framework and the LEDAPS product of Landsat TM. As shown in Sections 4 and 5, the non-atmospheric factors (e.g., the difference of RSRs of TM and MSS, the georegistration uncertainty, the radiometric calibration uncertainty, and image noises) bring more uncertainties than atmospheric factors (e.g., AOD and WV) overall. It proves that non-atmospheric factors cannot be ignored in time-series research using MSS data.

Considering the uncertainties brought by all the potential factors, the red and NIR-2 bands among all the MSS LSR bands and data acquired in spring (MAM), summer (JJA), and autumn (SON) among all the seasons are recommended for usage.

Additionally, the quality of MSS images varies considerably. The oversaturated pixels are shown in the QA band of MSS images, while some issues, including detector striping, are probably not shown in the QA band. In Sections 5, outliers in ROI 1 are found with unknown reasons, while some bands of the MSS images were found to be missing. The generated MSS LSR data need to be selected artificially for time-series analysis.

*6.3. Future Work*

The AODs retrieved by station-based visibility and satellite data have differences and inconsistencies. The station-based visibility records are obtained in spatial points with a specific time interval. At the same time, the image-based AOD is retrieved at the regional scale with different measure times from the visibility records. In the proposed framework, we use the most straightforward method, using the maximum value from a search radius of space and time for a scene of MSS images. Another method is to use spatial and temporal interpolation methods. The consistencies between AODs retrieved by ground-based visibility and satellite still require further research.

Another research direction is the evaluation. As the proposed framework needs land cover data from the 1970s as input, the evaluation is limited to China. However, the amount of Landsat MSS images covering China is unbalanced. The number of pairs of simultaneous images of MSS and TM used in evaluation became much lower than the total number of MSS and TM images, as the MSS and TM sensors were not always simultaneously turned on. The evaluation is not as thorough as the later Landsat 5, 7 [3], and Landsat 8 [4]. The study area will be extended to the whole world for future research.

Moreover, the usability of the MSS land surface reflectance product relates not only to atmospheric correction but also radiometric calibration, georegistration, denoising, cloud mask, and other factors. As the MSS LSR data are likely to be used in the time-series analysis together with other Landsat data, the consistency and correction of Landsat MSS and TM is also an important topic for future research. Though the work is largely promoted recently in radiometric calibration [5,6] and georegistration [8], more efforts are continuously needed to constitute the long-time-series Landsat MSS LSR archive.

## 7. Conclusions

The Landsat MSS dataset is the only multi-spectral global observation record by remote sensing in the 1970s, which plays a vital role in tracing time-series research to the 1970s. However, no MSS LSR product is currently publicly available. We propose a framework to generate the MSS LSR data using ground-based visibility records as input, overcoming the congenital deficiencies of the MSS sensor. The results are evaluated by the simultaneous observation by MSS and TM sensors in Landsat 4 and 5 using Landsat 4–5 TM LEDAPS product as the truth value. The evaluation results show that uncertainties decrease from the green band to the NIR band with a systemic bias. In addition, the non-atmospheric factors (e.g., the difference of RSRs of TM and MSS, the georegistration errors, the radiometric calibration uncertainty, and image noises) bring more uncertainties than atmospheric factors (e.g., AOD and WV) overall. Considering that both the sensors and preprocesses of MSS are comprehensively inferior to the later sensors and that the uncertainties brought by non-

atmospheric factors cannot be reduced by atmospheric correction, the evaluation results are comparable with the LEDAPS LSR product. We apply the MSS LSR data generated by the proposed framework on time series analysis in five ROIs of the spectral-stable land cover in China for all the MSS sensors. The application demonstrates the potential and promise of the MSS LSR data generated by the proposed framework. In general, the proposed framework can effectively generate the MSS LSR data of the 1970s.

**Author Contributions:** Conceptualization, C.Z., Z.W. and Q.Q.; Funding acquisition, Q.Q. and X.Y.; Investigation, C.Z.; Methodology, C.Z., Z.W. and Q.Q.; Project administration, Q.Q.; Resources, C.Z., Z.W., Q.Q. and X.Y.; Validation, C.Z. and Z.W.; Visualization, C.Z. and Z.W.; Writing—original draft, C.Z. and X.Y.; Writing—review editing, C.Z., Q.Q. and X.Y. All authors have read and agreed to the published version of the manuscript.

**Funding:** This work was supported by the National Natural Science Foundation of China (42071314) and China Postdoctoral Science Foundation (2021M690199).

**Institutional Review Board Statement:** Not applicable.

**Informed Consent Statement:** Not applicable.

**Data Availability Statement:** Not applicable.

**Acknowledgments:** The authors would like to thank USGS, NOAA, and NASA for the used data and Xu for the CNLUCC dataset.

**Conflicts of Interest:** The authors declare no conflict of interest.

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
