# Peer review of "A Framework of Generating Land Surface Reflectance of China Early Landsat MSS Images by Visibility Data and Its Evaluation"

_remotesensing, doi:10.3390/rs14081802_

Round 1

Reviewer 1 Report

I think the methodology used in the paper was validated, and it can be improved in future study, in the paper they used Collection1 data when Landsat introduces Collection3 data, the methodology could be refined and compared directly to the Landsat surface reflectance product.

Please see attached file for more comments.

Reviewer 2 Report

I have the following major issues with the manuscript:

  1. Why surface reflectance for the outdated dataset (MSS) is required?
  2. How the very coarse resolution NCEP data at 2.5 degrees is used for MSS data? Does it not produce uncertain final results?
  3. L185-187: How the correction factor was obtained?
  4. 2 is used to estimate AOD from visibility for a specific location, so how the AOD was estimated for other locations, where, visibility information is not available.
  5. Validation of the results is not convincing. I don’t understand the comparison of MSS data with TM and MODIS data because all the data used for validation are for a different period. To report the robustness of the proposed method, the author should implement this data on Landsat 8 OLI images and then compare their results with the LaSRC L8 surface reflectance product. If the results are comparable, then results for MSS data can be considered accurate.
  6. I have seen a new surface reflectance method published in 2019 in MDPI Remote Sensing which does not require AOD and other parameters for surface reflectance estimation. The authors should compare MSS results with the results MSS from that method.
  7. It is not recommended to use surface reflectance values as x10^3. The use of fractional values is recommended.

Reviewer 3 Report

A framework of generating land surface reflectance of China early Landsat MSS images by visibility data and its evaluation.

This paper proposed a framework for the Landsat MSS atmospheric correction based on station-obtained visibility records, independent of the remote sensing images.

The approach seems interesting to the readers of this journal. My recommendation would be acceptance of the article for publication following these minor corrections.

Line 12 and 17: In the summary section, explain the abbreviation or, if applicable, avoid them (AOD and LEDAPS).

Line 18: Keyword, do not repeat if it is in the title.

Line 66: Defina DDV.

Line 69: what is section 1?

Line 76-100: You could generate a table comparing some characteristics of the mentioned sensors.

Line 121-123: Please explain this sentence a bit more “center coordinates by a spatial and temporal search radius”.

Line 130: You could rename this section.

Line 173-177: Separate the units Example: “23km” “23 km”.

Reviewer 4 Report

The manuscript with the title “A framework of generating land surface reflectance of China early Landsat MSS images by visibility data and its evaluation.” proposes a framework to generate Landsat 1-5 MSS LSR data of China, using the records of ground-based visibility observation network as the input of 6S model. At first glance, the paper takes into a worthy topic; however, the structure is not strong. The manuscript fails to highlight the unique contribution compared to what is already available in the literature. This reviewer acknowledges the hard work made by the authors but also believes that the manuscript needs a lot of revisions/clarification and editions. 

Unfortunately, I can not recommend the manuscript suitable for publication in its present form. However, I suggest to the authors revise and resubmit the paper.

Here are my questions/revisions :

The first question that I need to ask is why not mention the LEDAPS methodology within the introduction section? LEDAPS generates Top of Atmosphere (TOA) Reflectance and TOA Brightness Temperature (BT) using the calibration parameters from the metadata. Auxiliary data such as water vapour, ozone, atmospheric pressure, Aerosol Optical Thickness (AOT), and digital elevation are then input with Landsat TOA Reflectance and TOA BT to Second Simulation of a Satellite Signal in the Solar Spectrum (6S) radiative transfer model to generate Surface Reflectance.

Later, the authors use LEDAPS products as “truth”, but no justification is provided or supported. Could you elaborate?

As the core of the paper, the methodology section should be described in an exact but also detailed manner. First, please add legends for the items used in the flowchart. For instance, what does represent the green, blue, and yellow boxes?    

The left side of Figure 3 is straightforward, MSS DN values-radiometric calibration-TOA reflectance (please note that radiance and reflectance are different things)-atmospheric corrections-Surface reflectance. The right side of Figure 3 is causing me some confusion; please revise the flow of the process and the direction of the arrows.

I suggest to breakdown Figure 3 and describing the methodology by parts.

Evaluation section.

Lines 201-206. What are the units of A, P, and U? Why use these metrics, why not other (e.g. RMSE)? Any reference?

Lines 130-133, This statement needs to be supported by references.

I feel like Section 5 is not part of this study. MODIS is not described in the Data section. The visual stability of the images is essential, but I do not see the purpose here. Figure 11 to 15 show visible differences in the median values, but what is the magnitude of these differences? The statistics from the tables do not say too much. I suggest revising this section.

No discussion? Why? There is a lot of room for discussion. For instance, what are the caveats of the proposed framework? Future research directions? Applications? Etc...

Conclusion sections should highlight the main finding of the manuscript based on the work done. As it is, it looks like a small discussion.

Round 2

Reviewer 4 Report

Many thanks to the authors for taking the time of addressing most of my revisions. After carefully reading, I am still not quite convinced of the improvement of Figure 3. A flowchart is, to my knowledge, a structure that follows a flow (or direction) and logic. For instance, it should be "input->"process"->"output".  The issue with Figure 3 is that the parameters( blue boxes according to the legend) are isolated.  I am assuming that they feed  6S Radiative transfer Model, but it is my assumption. Perhaps a second opinion on the flowchart would help.

Other than that, the manuscript's revisions, structure and flow look good.

Again, many thanks to the authors for taking the time and resubmit.